ecology, evolution, physiology

kleptoplast, fatty acid, reproduction, Sacoglossa

**Author for correspondence:**
Sónia Cruz
e-mail: sonia.cruz@ua.pt

## Photosynthesis from stolen chloroplasts can support sea slug reproductive fitness

Paulo Cartaxana[1,2], Felisa Rey[1,3], Charlotte LeKieffre[4], Diana Lopes[1], Cédric Hubas[5], Jorge E. Spangenberg[6], Stéphane Escrig[7], Bruno Jesus[8], Gonçalo Calado[9,10], Rosário Domingues[2,3], Michael Kühl[11], Ricardo Calado[1], Anders Meibom[7,12] and Sónia Cruz[1]

[1]CESAM—Centre for Environmental and Marine Studies, [2]Department of Biology, and [3]Mass Spectrometry Centre, LAQV-REQUIMTE, Department of Chemistry, University of Aveiro, Aveiro 3810-193, Portugal
[4]Cell and Plant Physiology Laboratory, University of Grenoble Alpes, CNRS, CEA, INRAE, Grenoble Cedex, France
[5]Biologie des Organismes et Écosystèmes Aquatiques (UMR BOREA 8067), Muséum National d'Histoire Naturelle, Sorbonne Université, Université de Caen Normandie, Université des Antilles, CNRS, IRD, Station Marine de Concarneau, Place de la croix, Concarneau 29900, France
[6]Institute of Earth Surface Dynamics (IDYST), University of Lausanne, Lausanne, CH-1015, Switzerland
[7]Laboratory for Biological Geochemistry, École Polytechnique Fédérale de Lausanne, Lausanne, CH-1015, Switzerland
[8]Laboratoire Mer Molécules Santé, Faculté des Sciences et des Techniques, Université de Nantes, Nantes 44322, France
[9]Department of Life Sciences, Lusófona University, Campo Grande 376, Lisbon 1749-024, Portugal
[10]NOVA School of Science and Technology, MARE—Marine and Environmental Sciences Centre, Campus de Caparica, Caparica 2829-516, Portugal
[11]Marine Biological Section, Department of Biology, University of Copenhagen, Strandpromenaden 5, Helsingør 3000, Denmark
[12]Center for Advanced Surface Analysis, Institute of Earth Sciences, University of Lausanne, Lausanne, CH-1015, Switzerland

PC, 0000-0001-5088-843X; CL, 0000-0002-9200-7925; CH, 0000-0002-9110-9292; JES, 0000-0001-8636-6414; BJ, 0000-0002-2047-3783; MK, 0000-0002-1792-4790; RC, 0000-0002-1670-9335; AM, 0000-0002-4542-2819; SC, 0000-0003-4775-8161

Some sea slugs are able to steal functional chloroplasts (kleptoplasts) from their algal food sources, but the role and relevance of photosynthesis to the animal host remain controversial. While some researchers claim that kleptoplasts are slowly digestible 'snacks', others advocate that they enhance the overall fitness of sea slugs much more profoundly. Our analysis shows light-dependent incorporation of $^{13}C$ and $^{15}N$ in the albumen gland and gonadal follicles of the sea slug *Elysia timida*, representing translocation of photosynthates to kleptoplast-free reproductive organs. Long-chain polyunsaturated fatty acids with reported roles in reproduction were produced in the sea slug cells using labelled precursors translocated from the kleptoplasts. Finally, we report reduced fecundity of *E. timida* by limiting kleptoplast photosynthesis. The present study indicates that photosynthesis enhances the reproductive fitness of kleptoplast-bearing sea slugs, confirming the biological relevance of this remarkable association between a metazoan and an algal-derived organelle.

## 1. Introduction

Sacoglossa is a group of sap-sucking sea slugs that feed on macroalgae. The most striking feature of some of these sea slugs is their ability to digest the algal cellular content while retaining intact functional chloroplasts (kleptoplasts) within the cells of their digestive gland [1,2]. This process of stealing plastids from algal cells (kleptoplasty) is more common in single-celled eukaryotes, such as foraminiferans, dinoflagellates and ciliates [3]. Recently, Van Steenkiste *et al.* [4] identified short-term functional kleptoplasts in two species of marine flatworms. However, among metazoans, the capacity for long-term maintenance (up to

several months) of functional chloroplasts remains a unique feature of a few species of sacoglossans [5,6]. Functional kleptoplasty occurs despite the absence of genetic material with an important role in chloroplast regulation, as these genes have been transferred to the algal nucleus over the evolution of endosymbiosis [7]. Although horizontal gene transfer was suggested as the primary reason underlying the long-term maintenance of photosynthesis in sacoglossan sea slugs, more recent studies found no evidence of genes supporting photosynthesis in the animal nuclear DNA [8].

The importance of kleptoplasty for the nutrition and metabolism of sacoglossan sea slugs remains controversial. Most studies have shown that photosynthesis plays an important role in individual survival and fitness over periods of food scarcity [9–12], while others argue that it is not essential for slugs to endure starvation [13]. Transcriptomic data on the sea slug *Elysia chlorotica* show that chloroplast sequestration leads to significant changes in host gene expression patterns throughout uptake and maturation, similar to that occurring during the establishment of symbiosis in corals, and suggest parallels between these animal–algal interactions [14].

Earlier radiolabelled carbon-based studies indicate the translocation of photosynthesis-derived metabolites from functional kleptoplasts into sacoglossan sea slug tissues [15–17]. Trench *et al.* [15] reported $^{14}$C-labelling within 2 h of incubation in the renopericardium, the cephalic neural tissue and the mucus-secreting pedal gland of *Elysia crispata* and *Elysia diomedea*. Recently, Cruz *et al.* [18] have shown initial light-dependent incorporation of $^{13}$C and $^{15}$N in digestive tubules followed by a rapid translocation and accumulation in kleptoplast-free organs of *Elysia viridis* (i.e. in tissues involved in reproductive functions such as the albumen gland and gonadal follicles). However, no direct relation between photosynthesis and reproductive investment of kleptoplast-bearing sea slugs has been established.

In the present study, we investigated the putative role of kleptoplast photosynthesis in the reproduction of the sacoglossan sea slug *Elysia timida* by (i) tracking short-term light-dependent incorporation of inorganic carbon and nitrogen into animal tissues using compound-specific isotope analysis (CSIA) of fatty acid methyl esters (FAMEs) and high-resolution secondary ion mass spectrometry (NanoSIMS), and (ii) investigating the effects of limiting photosynthesis (rearing animals under reduced light levels) in the number and fatty acid (FA) composition of spawned eggs. We report strong experimental evidence for a role of photosynthesis in the reproductive investment and fitness of a kleptoplast-bearing sea slug.

## 2. Results

### (a) Light-dependent incorporation of C and N
#### (i) NanoSIMS isotopic imaging
Semi-thin section imaging combined with NanoSIMS imaging showed that $^{13}$C- and $^{15}$N-labelling was not homogeneously distributed in different sea slug tissues, as $^{13}$C- and $^{15}$N-hotspots could be observed (figures 1–3). NanoSIMS images from individuals incubated in light for 6–36 h with $^{13}$C-bicarbonate and $^{15}$N-ammonium showed marked $^{13}$C- and $^{15}$N-labelling in kleptoplast-bearing digestive tubules (figure 1; electronic supplementary material, figure S1). Individuals incubated in the dark for 36 h displayed no $^{13}$C-

enrichment (electronic supplementary material, figure S1). By contrast, $^{15}$N-labelling was observed in the digestive tubules of sea slugs incubated in the dark for 36 h, although at a much lower level than in conspecifics incubated under light (electronic supplementary material, figure S1).

Marked $^{13}$C- and $^{15}$N-labelling was also observed in the albumen gland and the gonadal follicles (both kleptoplast free) of *E. timida* incubated in the light for 6–36 h with $^{13}$C-bicarbonate and $^{15}$N-ammonium (figures 2 and 3; electronic supplementary material, figure S1). $^{13}$C and $^{15}$N-labelling were still observed in the chasing phase, after individuals were transferred to fresh non-labelled artificial seawater (ASW) for up to another 12 h (figures 2 and 3; electronic supplementary material, figure S1). Again, no $^{13}$C-enrichment and lower $^{15}$N-labelling was observed in the albumen gland and the gonadal follicles of sea slugs incubated in the dark for 36 h, when compared to animals incubated in the presence of light (figures 2 and 3; electronic supplementary material, figure S1).

#### (ii) Fatty acid analysis
The most abundant FA (greater than 5% relative abundance) observed in *E. timida* were the saturated FA (SFA) 16 : 0 and 18 : 0, the monounsaturated FA (MUFA) 18 : 1$n$−9 and the polyunsaturated FA (PUFA) 18 : 2$n$−6, 18 : 4$n$−3, 20 : 4$n$−6, 20 : 5$n$−3, 22 : 4$n$−6 and 22 : 5$n$−3 (electronic supplementary material, table S1). In the presence of light, individuals incubated in $^{13}$C-bicarbonate enriched ASW for up to 36 h showed an increasing incorporation of $^{13}$C into FA over time (figure 4; electronic supplementary material, table S2). Incorporation of $^{13}$C occurred in all of the most abundant *E. timida* FA, except 18 : 4$n$−3 (figure 4). Levels of $^{13}$C-labelling decreased from SFA and MUFA precursors to longer-chain PUFAS. However, levels of incorporation of $^{13}$C in longer-chain FA 22 : 4$n$−6 and 22 : 5$n$−3 were higher than in 20 : 4$n$−6 and 20 : 5$n$−3, respectively (figure 4). In the chasing phase, when individuals were transferred to fresh non-labelled-ASW, levels of $^{13}$C in FA stopped increasing (figure 4). When animals were kept under dark conditions during 36 h of incubation with $^{13}$C-bicarbonate enriched ASW, sea slugs' FA showed no $^{13}$C-enrichment, with an incorporation equivalent to that of conspecifics incubated in the presence of light but in non-labelled-ASW (figure 4; electronic supplementary material, table S2).

*Acetabularia acetabulum* FA composition showed a lower diversity to that of *E. timida* (electronic supplementary material, table S1). Lower relative abundances of long-chain PUFA were found in the macroalgal tissue when compared to *E. timida*. PUFA 20 : 4$n$−6 was present in a relative abundance of 0.26% in the algae compared to 6.15% in the sea slug, while 22 : 4$n$−6, a major FA in *E. timida* (10.13%), was not present in *A. acetabulum*.

### (b) Effects of light treatment on egg masses
Pairs of *E. timida* initiated mating by meeting head-to-head and starting penis protrusion (electronic supplementary material, figure S2A). Animals mutually inserted their penis into the partner's female aperture, located at the base of the right parapodium. Spiral-shaped egg masses (electronic supplementary material, figure S2B) were spawned by *E. timida* in both light and quasi-dark conditions, although light treatment affected the number of spawning events. Sea slugs reared in regular light (40–160 µmol photons m$^{-2}$ s$^{-1}$) produced 7.5 ± 0.2 egg

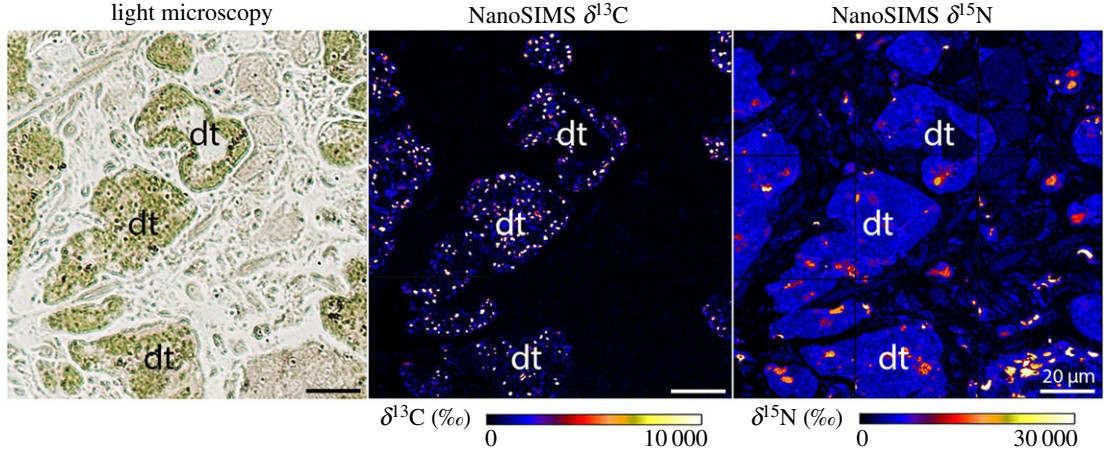

light microscopy NanoSIMS $\delta^{13}$C NanoSIMS $\delta^{15}$N

$\delta^{13}$C (‰) 0 10 000 $\delta^{15}$N (‰) 0 30 000

**Figure 1.** $^{13}$C and $^{15}$N incorporation in the digestive tubules of *Elysia timida*. Light microscopy picture and corresponding $\delta^{13}$C and $\delta^{15}$N NanoSIMS images of *E. timida* incubated in artificial seawater enriched with 2 mM NaH$^{13}$CO$_3$ and 20 µM$^{15}$NH$_4$Cl, for 6 h in the presence of light. Digestive tubules (dt) appear green on the light microscopy micrographs due to the presence of numerous chloroplasts. After 6 h of incubation strong $^{13}$C and $^{15}$N enrichment is observed in these structures. (Online version in colour.)

masses per pair (mean ± s.e.), corresponding to seven or eight egg masses over the 28-days experimental period. Spawning activity was more variable in pairs reared under reduced light levels (5 µmol photons m$^{-2}$ s$^{-1}$) and ranged from 1 to 5 egg masses per pair (3.0 ± 0.7, mean ± s.e.). The number of eggs produced by animals reared in regular light was significantly higher ($t_9 = 3.521$, $p = 0.007$) than for sea slugs reared under quasi-dark conditions (238 ± 13 versus 129 ± 30 eggs slug$^{-1}$ week$^{-1}$, respectively; figure 5). FA concentrations per egg were not significantly affected by light treatments (electronic supplementary material, figure S3 and table S3). FA composition was similar in *E. timida* individuals and egg masses (electronic supplementary material, tables S1 and S3).

## 3. Discussion

NanoSIMS isotopic imaging of the sea slug *E. timida* showed inorganic $^{13}$C incorporation in kleptoplast-bearing digestive tubules of light-exposed animals. However, light-dependent carbon incorporation was not restricted to these kleptoplast-bearing cells, and rapid accumulation (within 6 h) was observed in kleptoplast-free organs such as the albumen gland and gonadal follicles. $^{13}$C incorporation was not detected in the tissues of *E. timida* exposed to full darkness. Thus, our data clearly demonstrate that inorganic $^{13}$C was photosynthetically accumulated into functional kleptoplasts and subsequently translocated to other sea slug tissues, probably through soluble C-compounds (e.g. sugars) or FA.

Translocation of photosynthetically acquired carbon to animal tissues was previously identified in other species of sacoglossan sea slugs: *E. crispata*, *E. diomedea*, *E. viridis* and *Plakobranchus ocellatus* [15–18]. Using $^{14}$C, Trench *et al.* [15] observed labelling of kleptoplast-bearing digestive tubules after 15 min of light incubation for *E. crispata* and *E. diomedea*. Carbon incorporation was also detected in kleptoplast-free organs, such as the renopericardium (after 1 h), the cephalic neural tissue and the mucus-secreting pedal gland (after 2 h) and the intestine (after 5 h) [15]. Trench *et al.* [16] observed that $^{14}$C was incorporated into glucose and galactose in *E. viridis*, while Ireland & Scheuer [17] reported carbon incorporation in sugars and polypropionates for *P. ocellatus*. Using electron microscopy combined with NanoSIMS imaging, Cruz *et al.* [18] observed $^{13}$C-labelling after 1.5 h in starch

grains of kleptoplasts present in the kleptoplast-bearing digestive tubules of *E. viridis*, but $^{13}$C-labelling was also found in the cytoplasm surrounding the photosynthetic organelles. After longer incubation times (1.5–12 h), $^{13}$C-labelling was detected in *E. viridis* organs involved in reproduction, namely the albumen gland and gonadal follicles [18]. Evidence of fast translocation of photosynthates to kleptoplast-free animal tissues is not compatible with a previously proposed hypothesis that kleptoplasts are slowly digestible food reserves and that photosynthates produced are not continuously made available to the slug [13,19].

Sea slugs showed a much higher level of $^{15}$N enrichment in their tissues when incubated in the presence of light. Light-dependent incorporation of $^{15}$N was previously reported for *E. viridis* [18,20]. Teugels *et al.* [20] identified glutamine synthetase (GS)-glutamate synthetase (GOGAT) as the main pathway involved in N incorporation in the kleptoplasts. Hence, kleptoplasts may not only provide energy and carbon skeletons, but could also play a role in protein synthesis. De novo protein synthesis has been shown to occur for plastid-encoded membrane proteins in *E. chlorotica*, even after several months of starvation [21]. Contrary to $^{13}$C, our NanoSIMS imaging of *E. timida* recorded $^{15}$N incorporation in the dark. Nitrogen incorporated in specimens incubated in the dark (albeit significantly reduced) could result from the glutamine dehydrogenase (GDH) pathway in mitochondria [18,20].

The labelling with $^{13}$C was done in the absence of *A. acetabulum*, the macroalgal food source of *E. timida*, safeguarding that the labelled FA detected were not obtained heterotrophically (i.e. through grazing on *A. acetabulum*). Instead, $^{13}$C-labelled FA must have been synthesized in the kleptoplasts of the digestive tubules and eventually translocated to other animal cells. Additionally, labelled FA could have been produced in the animal cells through elongation/desaturation reactions using labelled precursors translocated from the kleptoplasts. In fact, the presence of labelled 22 : 4n−6 in *E. timida* is a direct evidence that the latter process occurred, as this FA was not present in *A. acetabulum*.

It was generally assumed that animals were unable to biosynthesize PUFA de novo since, presumably, they lacked specific desaturases required to produce 18 : 2n−6 (LA; linoleic acid) [22]. However, several findings have challenged this long-held assumption and it was recently shown that Δ

Proc. R. Soc. B 288: 20211779

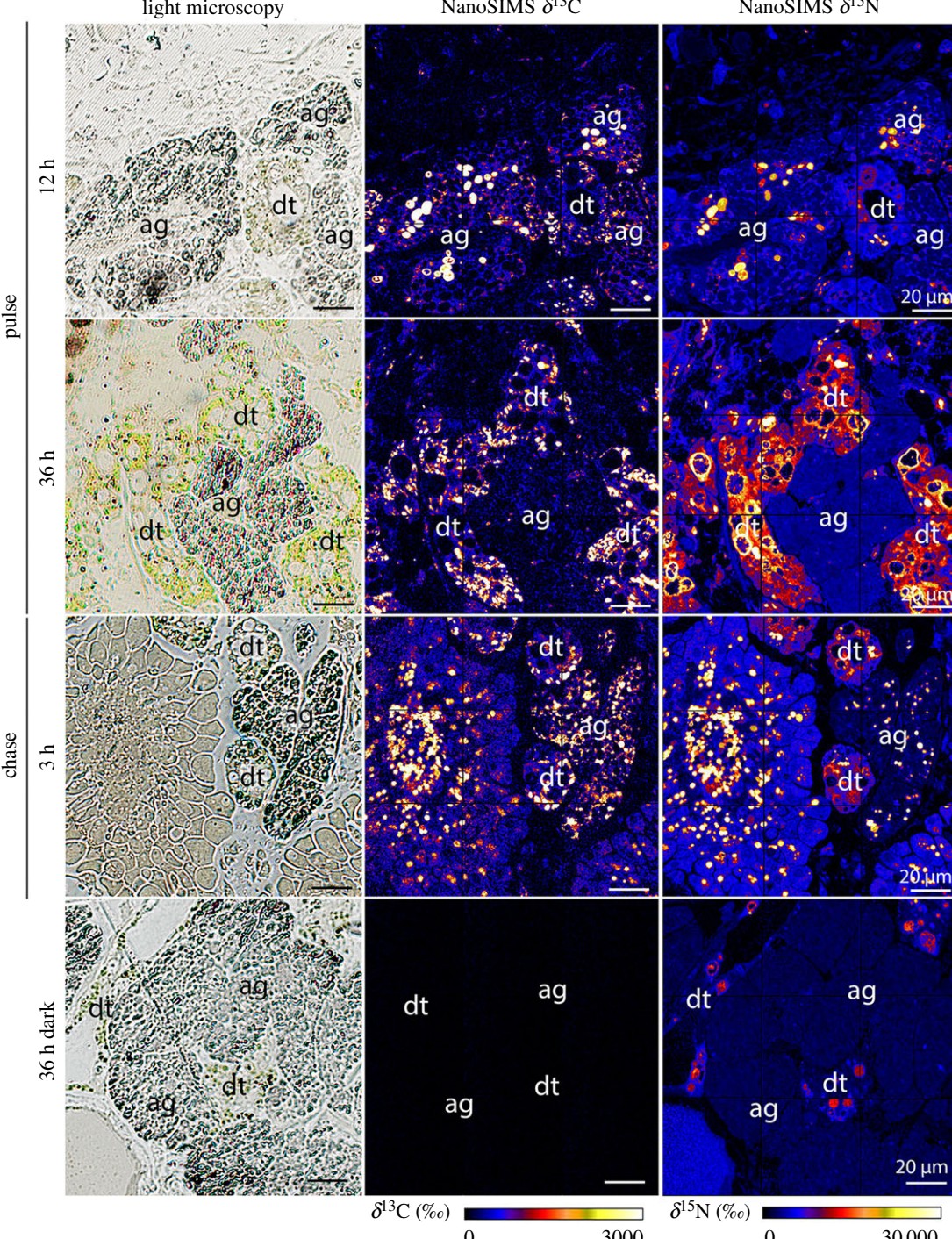

**Figure 2.** $^{13}$C and $^{15}$N incorporation in the albumen gland of *Elysia timida*. Light microscopy pictures and corresponding $\delta^{13}$C and $\delta^{15}$N NanoSIMS images of *E. timida* in an isotopic dual labelling pulse-chase experiment incubated in artificial seawater enriched with 2 mM NaH$^{13}$CO$_3$ and 20 µM $^{15}$NH$_4$Cl, in the presence of light for pulse (12 and 36 h) and chase (3 h), and in the dark for 36 h. ag, albumen gland; dt, digestive tubules. (Online version in colour.)

desaturases genes enabling de novo PUFA biosynthesis are widespread among invertebrates [23]. De novo biosynthesis of PUFA can occur via different pathways [24]. Tracking of $^{13}$C-labelled FA allowed us to infer the main pathway of PUFA biosynthesis in *E. timida* (electronic supplementary material, figure S4). A general dilution of the $^{13}$C signal was observed along the FA biosynthetic pathway from saturated and monounsaturated precursors 18 : 0 and 18 : 1$n$–9 to longer-chain PUFA. However, an increase in the $^{13}$C signal was observed in the last elongation steps of 22 : 5$n$–3 and 22 : 4$n$–6 production from 20 : 5$n$–3 (eicosapentaenoic acid, EPA) and 20 : 4$n$–6 (arachidonic acid; ARA), respectively. This finding indicates that the carbon donor (malonyl-CoA) during this elongation process

was $^{13}$C-enriched and thereby preferentially provided by kleptoplasts. Torres *et al.* [25] reported that methylmalonyl-CoA incorporating kleptoplast fixed-carbon is used by sacoglossan sea slugs in the synthesis of UV- and oxidation-blocking polypropionate pyrones by the action of FA synthase-like proteins. Pyrones could be critical for maintaining long-term photosynthetic activity in sacoglossan sea slugs by serving antioxidant and photoprotective roles [17,25].

Dietary PUFA have been shown to modulate marine invertebrate gametogenesis, embryogenesis and larval development [26–28]. The levels of PUFA recorded in female tissues and embryos of the sea snail *Crepidula fornicata* were related to its reproductive output [29]. Bautista-Teruel *et al.* [30] reported

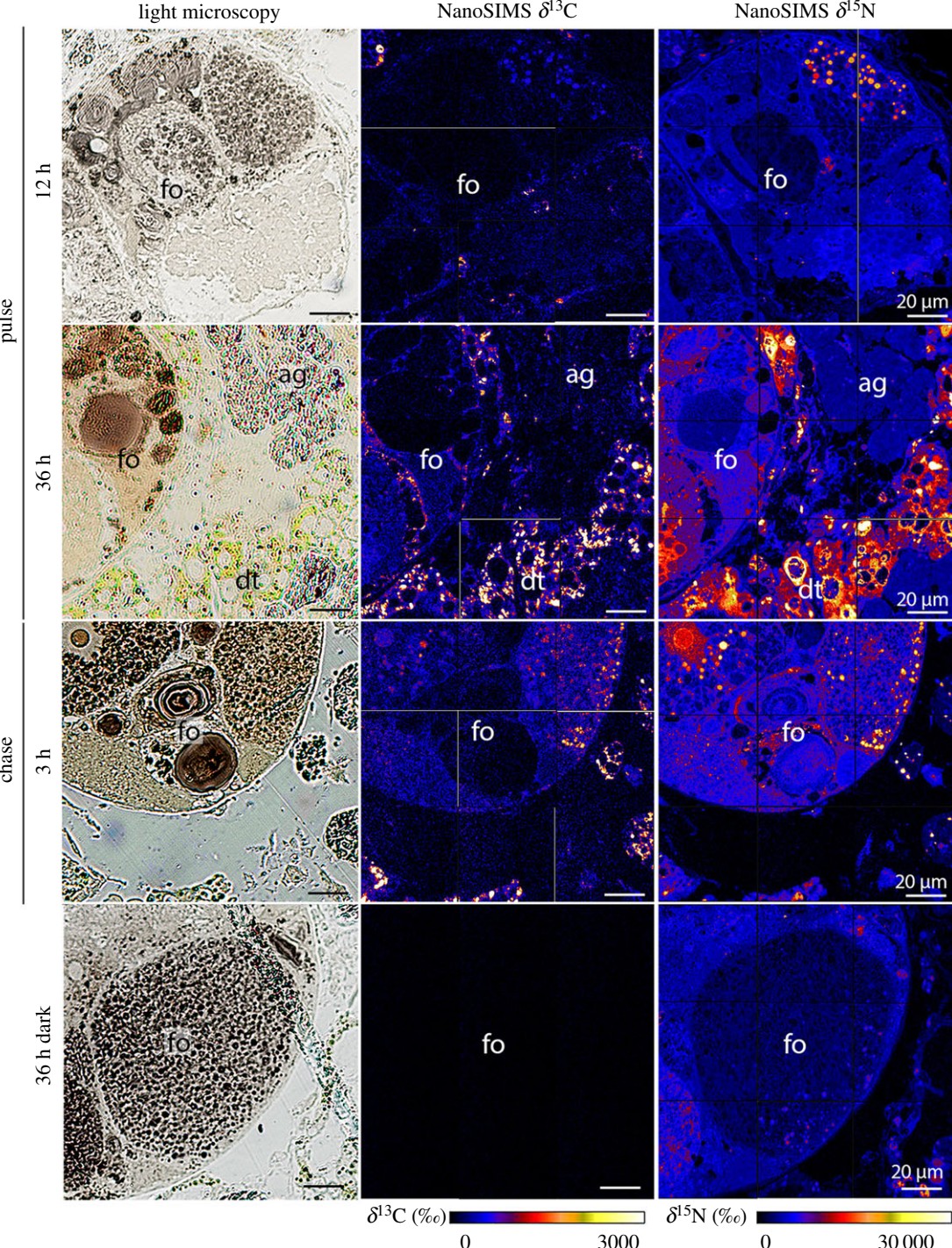

**Figure 3.** $^{13}$C and $^{15}$N incorporation in the gonadal follicles of *Elysia timida*. Light microscopy pictures and corresponding $\delta^{13}$C and $\delta^{15}$N NanoSIMS images of *E. timida* in an isotopic dual labelling pulse-chase experiment incubated in artificial seawater enriched with 2 mM NaH$^{13}$CO$_3$ and 20 µM $^{15}$NH$_4$Cl, in the presence of light for pulse (12 and 36 h) and chase (3 h), and in the dark for 36 h. fo, gonadal follicles; dt, digestive tubules; ag, albumen gland. (Online version in colour.)

that the reproductive performance in the gastropod *Haliotis asinina* was linked to diets with increased levels of PUFA, such as EPA and ARA. The latter FAs are precursors of prostaglandins, a group of biologically active compounds participating in marine invertebrate reproduction [31,32]. Hence, the assembly of photosynthesis-driven long-chain PUFA as shown in our study for *E. timida* (in the kleptoplasts and elsewhere in animal cells using photosynthesis-derived precursors) are likely to play a crucial role in the reproductive output of this species and increase evolutionary fitness.

Fecundity, assessed as the number of eggs spawned by *E. timida* along a four week-period, was significantly higher in sea slugs exposed to regular light (40–160 µmol photons m$^{-2}$ s$^{-1}$) than in animals reared under reduced light (5 µmol photons m$^{-2}$ s$^{-1}$). Under a period of resource shortage (i.e. limited photosynthesis under reduced light levels), *E. timida* clearly changed its reproductive energy investment by decreasing the number of spawned egg masses. Shiroyama *et al*. [33] observed higher number of eggs spawned by *Elysia atroviridis* when fed under regular light (30 µmol photons m$^{-2}$ s$^{-1}$) than when kept under reduced light conditions (1 µmol photons m$^{-2}$ s$^{-1}$).

In our study, complete darkness was not used to inhibit photosynthesis in order not to disrupt diel biorhythms and animal behaviour. The sea slug *E. viridis* was reported to become inactive under full darkness, rarely seeming to feed

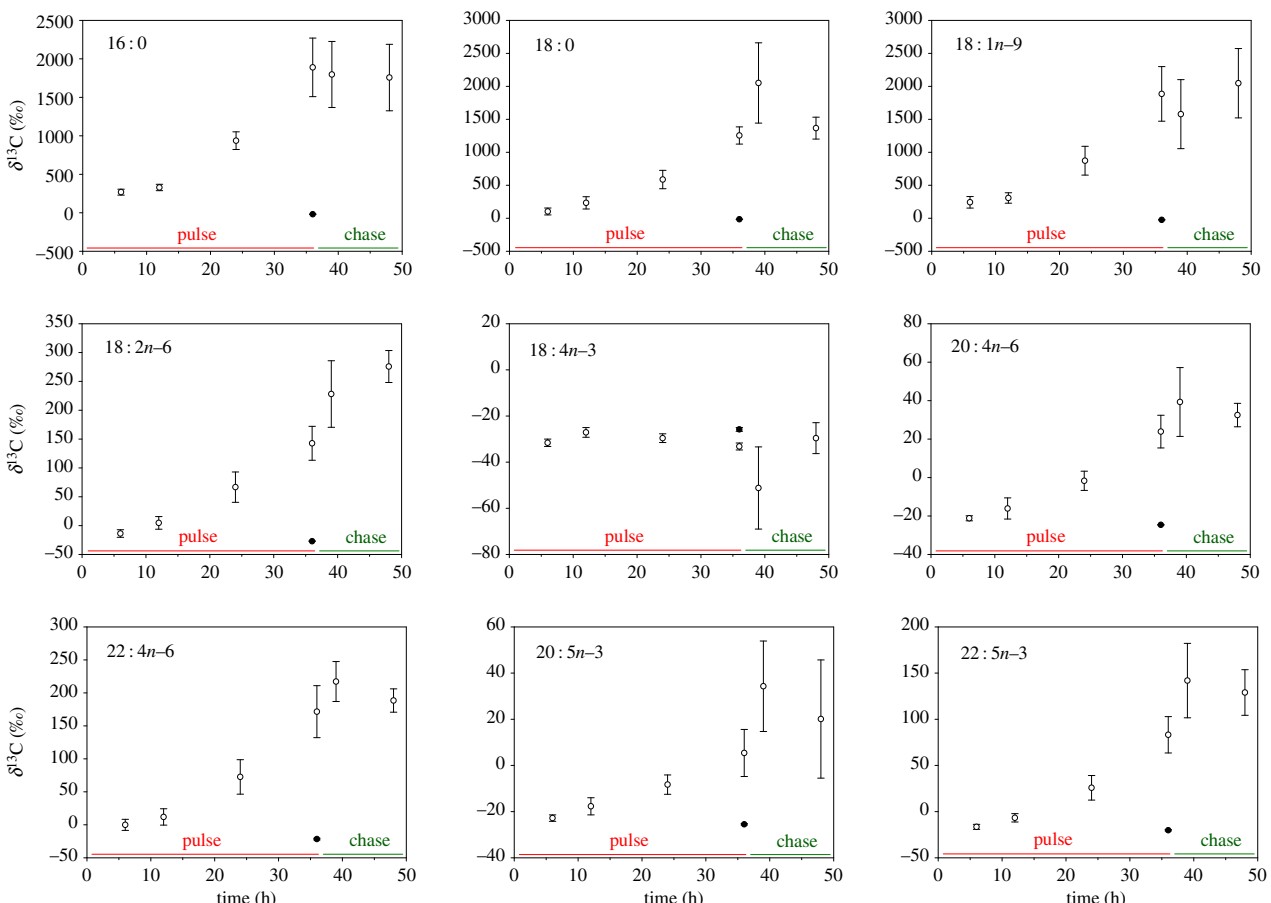

**Figure 4.** $^{13}$C incorporation in the main fatty acids of *Elysia timida*. $^{13}$C (‰) in most abundant fatty acids of *E. timida* as a function of time (h) in an isotopic labelling pulse-chase experiment in artificial seawater enriched with 2 mM NaH$^{13}$CO$_3$ in the presence of light (open circles) or in dark-incubated specimens for 36 h (closed circles). Mean ± s.e., $n = 3$. (Online version in colour.)

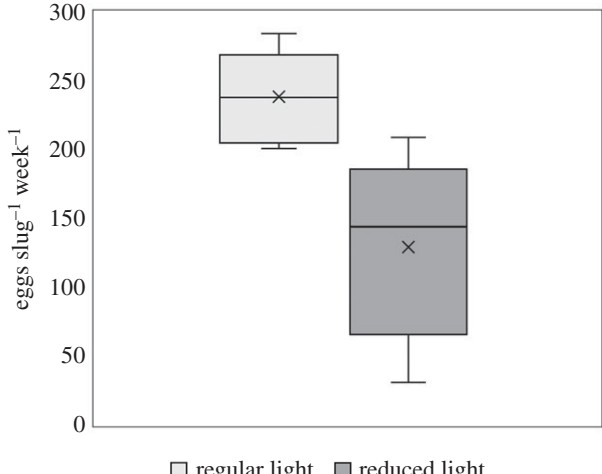

**Figure 5.** Fecundity of *Elysia timida*. Number of eggs spawned by *E. timida* exposed to a 14 : 10 h light/dark photoperiod and a scalar irradiance of 40–160 µmol photons m$^{-2}$ s$^{-1}$ (regular light) or 5 µmol photons m$^{-2}$ s$^{-1}$ (reduced light) for 28 days. The line is the median, the *x* represents the mean, top and bottom of the box are the 75% and 25% percentile, and the whiskers represent the maximum and minimum values. Animals were fed continuously with *Acetabularia acetabulum*. Differences between treatments were significant at $p = 0.007$.

[9]. Hence, an extremely low light intensity, enough to substantially limit kleptoplast photosynthesis, was used instead of full darkness [34]. Animals in both light treatments were

observed attached to the macroalgae, and the cellular content of *A. acetabulum* was emptied similarly in both light treatments. This indicates that the heterotrophic feeding ability of sea slugs was not affected by the experimental conditions. Although we cannot rule out a direct effect of experimental light conditions in sea slug spawning, we hypothesize that observed differences in fecundity were due to resources provided by kleptoplast photosynthesis. While reproductive investment was reduced when photosynthesis was limited, *E. timida* allocated similar amounts of FA to individual eggs regardless of light treatment. This is particularly relevant for egg viability in species such as *E. timida*, in which offspring success depends exclusively on the parental provisioning of endogenous reserves to fuel embryonic development and early larval life, until lecithotrophic larvae are able to metamorphose to the imago of the adult and feed on exogenous food sources [35].

In conclusion, we report the allocation of photosynthates to kleptoplast-free organs involved in the reproduction of *E. timida*, along with photosynthesis-driven assembly of long-chain PUFA, and higher sea slug fecundity under regular light than in quasi-dark conditions. These results indicate that kleptoplast photosynthesis increases the reproductive investment of *E. timida*. It has been shown that kleptoplasty in Sacoglossa contributes to survival and fitness in periods of food scarcity, in some cases allowing individuals to endure several months of starvation [9,10]. We hypothesize that functional kleptoplasty in sacoglossan sea slugs may further potentiate species' success by maximizing its reproductive output.

# 4. Material and methods

## (a) Animal collection and maintenance

Specimens of *E. timida* (Risso, 1818) (electronic supplementary material, figure S2A) were collected in Puerto de Mazarrón in the Mediterranean Sea, Spain. Sampling of *E. timida* and its algal food, *A. acetabulum* (Linnaeus) P. C. Silva, 1952, was done by scuba diving at a depth of approximately 2 m. Animals and macroalgae were kept in aerated seawater collected at the sampling site and transported to the laboratory within 48 h. Sea slugs and macroalgae were maintained for two weeks in a 150 l recirculated life support system (LSS) operated with ASW at 18°C and a salinity of 35. The photoperiod was kept at 14 h light : 10 h dark, with a photon scalar irradiance of 60 µmol photons $m^{-2}\,s^{-1}$ being provided by T5 fluorescent lamps. Photon scalar irradiance was measured with a Spherical Micro Quantum Sensor and a ULM-500 Universal Light Meter (Heinz Walz GmbH, Germany). The laboratory adaptation period was chosen to ensure replicability in feeding and the light history of the animals at the beginning of the experiment.

## (b) Light-dependent incorporation of C and N

### (i) Dual isotopic labelling incubations

Isotopic dual labelling pulse-chase experiments were conducted in closed-systems (1 l glass bottles, three independent containers per treatment). Labelled-ASW was prepared in accordance with Harrison *et al.* [36], but using $NaH^{13}CO_3$ ($^{13}C$ isotopic abundance of 99%, Sigma-Aldrich) and $^{15}NH_4Cl$ ($^{15}N$ isotopic abundance of 98%, Sigma-Aldrich) to a final concentration of 2 mM and 20 µM, respectively (labelled-ASW). Non-labelled-ASW (control-ASW) contained $NaHCO_3$ and $NH_4Cl$ (Sigma-Aldrich) in the same concentrations as the isotopically enriched ASW. Sea slugs were incubated in labelled- and control-ASW at 18°C, in the absence of their food source, and under a photon scalar irradiance of 100 µmol photons $m^{-2}\,s^{-1}$ (measured inside the glass bottles). Additionally, sea slugs in labelled-ASW were incubated in full darkness (electronic supplementary material, table S4). Dark conditions served as a control for light-independent carbon and nitrogen incorporation. The pulse of isotopic dual labelling started 1 h after the onset of the light period. A subset of three individuals kept in labelled-ASW and exposed to light were sampled after 6, 12, 24 and 36 h of incubation (pulse phase), quickly rinsed with distilled water, flash frozen in liquid nitrogen and stored at –80°C until further FA analysis. An additional individual kept in labelled-ASW and exposed to light was collected at each of the referred time points, rinsed and fixed in 0.2 M cacodylate buffer containing 4% glutaraldehyde and 0.5 M sucrose and stored at 4°C for 24 h before tissue preparation for secondary ion mass spectrometry imaging (NanoSIMS 50 l). Individuals from labelled-ASW incubated in dark conditions and from control-ASW exposed to light were sampled after 36 h as described above for FA and SIMS analysis. The remaining individuals in labelled-ASW and exposed to light were transferred to fresh control-ASW. During this chase period, a subset of individuals was collected after 3 and 12 h as described above, for FA and SIMS analysis (electronic supplementary material, table S4). In light treatments, the light was set constant throughout the 48 h experiment.

### (ii) High-resolution secondary ion mass spectrometry (NanoSIMS) isotopic imaging

Tissue preparation for NanoSIMS imaging was done according to Cruz *et al.* [18] (see electronic supplementary material, method S1). Large areas of interest were imaged with a Nano-SIMS 50 L secondary ion mass spectrometer. This allowed imaging of the subcellular distribution of $^{13}C$ and $^{15}N$ enrichment in the exact same areas of the imaged histological overviews described above, enabling a direct correlation of structural and isotopic images. All measurements were performed using the following analytical conditions: 16 keV primary ion beam of $Cs^+$ focused to a beam spot of *ca* 100–150 nm and counting $^{12}C_2^-$, $^{13}C^{12}C^-$, $^{14}N^{12}C^-$ and $^{15}N^{12}C^-$ ions in electron multipliers at a mass resolution of greater than 8000 (Cameca definition), enough to resolve potential interferences in the mass spectra. Images captured with NanoSIMS 50 l were processed using the L'IMAGE software (Larry R. Nittler, Carnegie Institution of Washington, Washington DC, USA). Regions of interest selecting individual anatomic structures were defined, and distribution maps of $^{13}C/^{12}C$ and $^{15}N/^{14}N$ ratios were obtained by taking the ratio between the drift-corrected $^{13}C^{12}C^-$ and $^{12}C^{12}C^-$ images, and $^{15}N^{12}C^-$ and $^{14}N^{12}C^-$ images, respectively. Five stacked planes were used for each image. $^{13}C$ and $^{15}N$ enrichment values in the figures were expressed as delta notations, $\delta^{13}C = (C_{mes}/C_{nat} - 1) \times 1000$ and $\delta^{15}N = (N_{mes}/N_{nat} - 1) \times 1000$, where $C_{mes}$ and $N_{mes}$ are the measured $^{12}C^{13}C^-/^{12}C_2^-$ and $^{15}N^{12}C^-/^{14}N^{12}C^-$ ratios of the sample and $C_{nat}$ and $N_{nat}$ is the average $^{12}C^{13}C^-/^{12}C_2^-$ and $^{15}N^{12}C^-/^{14}N^{12}C^-$ ratios measured in control, non-labelled samples. A number of measurements on these controls ($n = 12$) yielded distributions of $\delta^{13}C = 0 \pm 9.6‰$, and $\delta^{15}N = 0 \pm 21.6‰$ ($\pm 2\sigma$).

### (iii) Compound-specific isotope analysis of fatty acid methyl esters

Fatty acid extraction was performed following the method of Bligh & Dyer [37] as modified by Meziane & Tsuchiya [38] and Passarelli *et al.* [39] (see electronic supplementary material, method S2). The CSIA of the FAME was performed by gas chromatograph/combustion/isotope ratio mass spectrometry (GC/C/IRMS) with an Agilent 6890 GC instrument coupled to a Thermo Fisher Scientific (Bremen, Germany) Delta V Plus IRMS instrument via a combustion interface III under a continuous helium flow. The GC separation was performed with the HP-FFAP column (50 m × 0.20 mm; length × inner diameter) coated with 0.33 µm nitroterephthalic acid modified polyethylene glycol stationary phase. The FAME samples were injected splitless at 230°C. After an initial period of 2 min at 100°C, the column was heated to 240°C (held 26 min) at 5°C $min^{-1}$, then to 245°C (held 4 min). This GC conditions were optimized for good separation of unsaturated FAs by injection of a standard mixture of 37 FAMEs (Supelco 37 Component FAME Mix, Sigma-Aldrich, Buchs, Switzerland) containing C4–C24 homologues. For calibration and normalization of the measured FAME $\delta^{13}C$ values were used the previously determined $\delta^{13}C$ values (by elemental analysis/IRMS) of a mixture of deuterated carboxylic acids used as external standards. For quality control, the repeatability and intermediate precision of the GC/C/IRMS analysis and the performance of the GC and combustion interface were evaluated every five runs by injection of a carefully prepared mixture of FAMEs reference materials [40]. The standard deviation for repeatability of the $\delta^{13}C$ values ranged between ±0.05 and ±0.5‰ for *m/z* 45 peak size between 15000 mV and less than 500 mV. The FA $\delta^{13}C$ were determined from the FAME $\delta^{13}C$ by correction for the isotopic shift due to the carbon introduced by methylation using a mass balance equation [41].

## (c) Effects of light treatment on egg masses

A floating tray with wells (56 mm diameter × 60 mm depth) was placed floating in the described LSS. The bottom of the wells was made of a 0.5 mm mesh to allow water exchange [42]. A re-circulating water pump was placed below the experimental tray to increase water renewal inside the wells. Twenty-four adult *E. timida* specimens were randomly divided in pairs and placed in individual wells. The photoperiod was kept at 14 h light : 10 h

dark. Two treatments (six replicates per treatment, each replicate being a pair of sea slugs) were performed: (i) 'regular light' treatment, in which the sea slug specimens were subjected to a photon scalar irradiance of 40–160 µmol photons $m^{-2} s^{-1}$, depending on the position inside the well; (ii) 'reduced light', in which the sea slug specimens were subjected to a photon scalar irradiance of 5 µmol photons $m^{-2} s^{-1}$. Light treatments were achieved by placing either transparent or opaque lids over the wells. In the case of the reduced irradiance treatment, light reached the animals through the bottom mesh. Animals were fed every day with *A. acetabulum* grown at a photon scalar irradiance of 60 µmol photons $m^{-2} s^{-1}$ under a 14 h light : 10 h dark photoperiod. During the experimental period, one animal died in the reduced light conditions, reducing the number of replicates in this treatment to $n = 5$.

Egg masses spawned by the sea slugs on the walls of the wells and, occasionally, on the net at the bottom of the wells were counted daily for 28 days and collected using a scalpel (electronic supplementary material, figure S2B). The number of eggs in each individual egg mass was counted using a Leica DMS300 digital microscope. Egg masses were gently washed in ultrapure water, frozen at −80°C and freeze-dried. The last egg mass produced in each experimental unit (well) was analysed for FA composition. Five replicates of *E. timida* sea slugs reared in actinic light conditions and *A. acetabulum* were similarly washed in ultrapure water, frozen at −80°C, freeze-dried and macerated prior to lipid extraction. After lipid extraction (see electronic supplementary material, method S3), FAs in the three biological matrices surveyed (egg masses, sea slugs and macroalgae) were transmethylated according to Aued-Pimentel *et al.* [43] to obtain FAME and analysed by gas chromatography—mass spectrometry (GC-MS). FAME identification was performed by comparing retention times and mass spectra with those of commercial FAME standards (Supelco 37 Component FAME Mix, ref. 47885-U, Sigma-Aldrich) and confirmed by comparison with the Wiley library and the spectral library from 'The Lipid Web' [44]. FA quantification was performed using calibration curves obtained from FAME standards under the same instrumental conditions. FA in *E. timida* and *A. acetabulum* were expressed as relative abundances (%). FA concentrations in the eggs were expressed as pg $egg^{-1}$ dividing the FA content of the whole egg mass by the number of eggs.

## (d) Statistical analyses

The number of eggs spawned in each experimental unit (pairs of sea slugs placed on each well) was averaged to avoid pseudo-replication, and averages were treated as independent replicates [45]. Statistically significant differences in the number and FA concentrations of eggs spawned by regular versus reduced light reared animals were tested using independent samples *t*-tests. Normality was checked using a Shapiro–Wilk's test, while homogeneity of variances was tested using Levene's test. Statistical analyses were carried out using IBM SPSS Statistics 24.

Data accessibility. Raw data are available from the Dryad Digital Repository: https://doi.org/10.5061/dryad.612jm643q [46]. Supplementary information is provided in the electronic supplementary material [47].

Authors' contributions. P.C.: Conceptualization, formal analysis, investigation, writing—original draft, writing—review and editing; F.R.: formal analysis, investigation, writing—review and editing; C.L.: formal analysis, investigation, writing—review and editing; D.L.: formal analysis, investigation, writing—review and editing; C.H.: conceptualization, formal analysis, investigation, writing—review and editing; J.E.S.: formal analysis, investigation, methodology, writing—review and editing; S.E.: formal analysis, investigation, writing—review and editing; B.J.: conceptualization, formal analysis, investigation, writing—review and editing; G.C.: conceptualization, formal analysis, writing—review and editing; R.D.: methodology, writing—review and editing; M.K.: conceptualization, funding acquisition, writing—review and editing; R.C.: conceptualization, writing—review and editing; A.M.: conceptualization, funding acquisition, methodology, writing—review and editing; S.C.: conceptualization, formal analysis, funding acquisition, investigation, methodology, project administration, supervision, writing—original draft, writing—review and editing. All authors gave final approval for publication and agreed to be held accountable for the work performed therein.

Competing interests. The authors declare no competing interests.

Funding. This project has received funding from the European Research Council (ERC) under the European Union's Horizon 2020 research and innovation programme (grant agreement no. 949880) (S.C.). Other funding: Fundação para a Ciência e a Tecnologia (FCT/MCTES), grant no. 2020.03278.CEECIND (S.C.); FCT/MCTES, grant CEECIND/01434/2018 (P.C.); FCT/MCTES, grant no. CEECIND/00580/2017 (F.R.); Gordon and Betty Moore Foundation, grant no. GBMF9206 (M.K.); Swiss National Science Foundation, grant 200021_179092 (A.M.); FCT/MCTES, grant no. UIDB/50017/2020+UIDP/50017/2020; FCT/MCTES, grant no. UIDB/04292/2020.

Acknowledgements. We thank Dr José Templado and Dr Marta Calvo for help in the collection of *E. timida* and *A. acetabulum* and Sofie Jakobsen and Gabriel Ferreira for technical assistance.

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
