## [Peer Review File · Proceedings of the Royal Society B: Biological Sciences]

Review History

RSPB-2021-1331.R0 (Original submission)

Review form: Reviewer 1

Recommendation

Accept as is

Scientific importance: Is the manuscript an original and important contribution to its field?

Excellent

General interest: Is the paper of sufficient general interest?

Excellent

Quality of the paper: Is the overall quality of the paper suitable?

Excellent

Is the length of the paper justified?

Yes

Should the paper be seen by a specialist statistical reviewer?

No

Do you have any concerns about statistical analyses in this paper? If so, please specify them explicitly in your report.

No

It is a condition of publication that authors make their supporting data, code and materials available - either as supplementary material or hosted in an external repository. Please rate, if applicable, the supporting data on the following criteria.

Is it accessible?

Yes

Is it clear?

Yes

Is it adequate?

Yes

Do you have any ethical concerns with this paper?

No

Comments to the Author

Review of Manuscript ID: RSPB-2021-1331 Photosynthesis from stolen chloroplasts increases sea slug reproductive fitness

Summary: This is a strong paper which details careful experiments to demonstrate that the sea slug *Elysia timida* derives significant nutritional benefit from its kleptoplastic association with chloroplasts stolen from *Acetabularia acetabulum*. The authors use ^{13}C and ^{15}N labelling to show that organic molecules produced by photosynthesis are incorporated into the sea slug tissues, and translocated to the albumen gland and gonadal follicles (which are NOT located near chloroplasts). The use of NanoSIMS isotopic imaging was particularly useful in demonstrating this transfer. They also measured the amount and types of fatty acids, included those reported to be used in reproduction, in slugs using ^{13}C labelling under light and dark conditions. In addition, they demonstrate that eggs produced by sea slugs under actinic light lay statistically significantly more eggs than those kept in non-actinic light. Taken together, these three lines of analysis strongly support the hypothesis that kleptoplasty provides nutrition for sea slugs, and improves their overall evolutionary fitness.

The author's major conclusions are:

1. Sequestered chloroplasts in *E. timida* actively photosynthesize fixing inorganic carbon to organic molecules which are then translocated to non-photosynthetic tissues in the animal involved with reproduction
2. Radiolabelled carbon was shown to be incorporated in a range of fatty acids, including those associated with reproduction, in animals exposed to light only. This experiment also provided an opportunity to trace the biochemistry of fatty acid synthesis in the animal.
3. Egg fecundity was statistically higher in slugs reared under actinic conditions vs non-actinic.

Comments: This is a very well written manuscript. The experiments are well designed, well executed, and conclusively support the hypothesis that *E. timida* rely on photosynthesis from their stolen algal chloroplasts. While there is significant data in the literature supporting this position, recent papers have cast doubt on the role of photosynthesis. In my opinion, this study is of particular importance because it is the first to prove the link between photosynthesis and actual reproductive fitness in the animals. I recommend this manuscript for publication and only have a few minor comments.

Minor comments:

Line 121: You state that the ^{13}C level "generally leveled out". Could you clarify what this means? As in the rate of incorporation leveled out, or the total amount leveled out indicating no new incorporation of ^{13}C ?

Figures: In general, I think more detailed descriptions of the figures would be helpful.

For example:

Fig 1: Those less adept at microscopy or slug anatomy may not be able to determine where the ^{13}C and ^{15}N are localized. It would be helpful to clarify in the figure text what is going on in the pictures.

Fig 2: The figures have the albumen glands labelled, but why not labels for the digestive tubule?

Fig 3: What is the tissue on the left hand side of the 3H chase micrograph? Are those follicles?

Again, why not label more regions for clarity?

Review form: Reviewer 2

Recommendation

Major revision is needed (please make suggestions in comments)

Scientific importance: Is the manuscript an original and important contribution to its field?

Good

General interest: Is the paper of sufficient general interest?

Excellent

Quality of the paper: Is the overall quality of the paper suitable?

Good

Is the length of the paper justified?

Yes

Should the paper be seen by a specialist statistical reviewer?

No

Do you have any concerns about statistical analyses in this paper? If so, please specify them explicitly in your report.

No

It is a condition of publication that authors make their supporting data, code and materials available - either as supplementary material or hosted in an external repository. Please rate, if applicable, the supporting data on the following criteria.

Is it accessible?

Yes

Is it clear?

Yes

Is it adequate?

Yes

Do you have any ethical concerns with this paper?

No

Comments to the Author

Several papers have demonstrated that kleptoplasts in the sea slug provide photosynthate and support growth, but recent study has denied it. Therefore, the role of kleptoplasts in the sea slug is still controversial. This study demonstrated that photosynthate from kleptoplasts are

translocated to kleptoplasts-free reproductive organs and it might enhance the reproductive fitness. Authors showed the translocation of photosynthate by tracking short-term light-dependent incorporation of inorganic carbon (^{13}C) and nitrogen (^{15}N) into animal tissues using compound specific isotope analysis (CSIA) of fatty acid methyl esters (FAME) and high-resolution secondary ion mass spectrometry (NanoSIMS). This reviewer feels the experimental design is very reasonable and results are clear for this portion of the paper. However, authors also showed the physiological significance of the photosynthate from kleptoplasts by testing the effect of light on fecundity of sea slug. This reviewer feels the conclusion that the photosynthate from kleptoplasts is important for the reproductive fitness of the sea slug is not supported by the current data set. This reviewer feels the data reported in this paper is important to understanding the role of the kleptoplasts in sea slugs, however, in order to recommend publication either additional data needs to be added or the discussion/conclusion needs to be modified.

Major comment

In order to conclude that photosynthesis enhances the reproductive fitness of the sea slug, photosynthesis needed to be directly and specifically suppressed using an inhibitor such as DCMU, rather than reduced light intensity. The authors cannot exclude that reduced light intensity may directly reduce the fecundity of the sea slugs. Author need to add new data to support their conclusion. Alternatively, authors need to change this conclusion to discussion. Furthermore, throughout the text (and the title), it should be changed to say reduced light intensity and not inhibited/suppressed photosynthesis.

Minor comment

L62-65, It might be important to say that there is no horizontal gene transfer from macroalgae to sea slug (Maeda et al. 2021 eLife).

Decision letter (RSPB-2021-1331.R0)

29-Jul-2021

Dear Dr Cruz:

I am writing to inform you that your manuscript RSPB-2021-1331 entitled "Photosynthesis from stolen chloroplasts increases sea slug reproductive fitness" has, in its current form, been rejected for publication in Proceedings B.

This action has been taken on the advice of referees, who have recommended that substantial revisions are necessary. With this in mind we would be happy to consider a resubmission, provided the comments of the referees are fully addressed. However please note that this is not a provisional acceptance.

- 1) A 'response to referees' document including details of how you have responded to the comments, and the adjustments you have made.

- 2) A clean copy of the manuscript and one with 'tracked changes' indicating your 'response to referees' comments document.
- 3) Line numbers in your main document.
- 4) Data - please see our policies on data sharing to ensure that you are complying (<https://royalsociety.org/journals/authors/author-guidelines/#data>).

Sincerely,
 Dr Daniel Costa
 mailto: proceedingsb@royalsociety.org

Associate Editor

Comments to Author:

Thank you for submitting your manuscript entitled "Photosynthesis from stolen chloroplasts increases sea slug reproductive fitness" to Proceedings B. We have gotten back the reviews and both are generally positive. Both reviewers agree that the authors have strongly demonstrated that the sea slug makes use of and obtains nutritional benefits from kleptoplasts, and this contributes to the controversy around this topic. The demonstration of translocation of the photosynthates to reproductive cells also indicate potential enhancement of reproductive fitness. However, Reviewer 2 brings up an important concern which can weaken the conclusion that kleptoplast photosynthesis does enhance reproductive fitness. Reviewer 2 notes that the methodology used for this aspect of the study, in particular the different light treatments, can also have a direct impact on the fecundity of the sea slugs and cannot be said to necessarily inhibit or suppress photosynthesis. Reviewer 2 provides suggestions to address the comments.

Reviewer(s)' Comments to Author:

Referee: 1

Comments to the Author(s)

Review of Manuscript ID: RSPB-2021-1331 Photosynthesis from stolen chloroplasts increases sea slug reproductive fitness

Summary: This is a strong paper which details careful experiments to demonstrate that the sea slug *Elysia timida* derives significant nutritional benefit from its kleptoplastic association with chloroplasts stolen from *Acetabularia acetabulum*. The authors use ¹³C and ¹⁵N labelling to show that organic molecules produced by photosynthesis are incorporated into the sea slug tissues, and translocated to the albumen gland and gonadal follicles (which are NOT located near chloroplasts). The use of NanoSIMS isotopic imaging was particularly useful in demonstrating this transfer. They also measured the amount and types of fatty acids, included those reported to be used in reproduction, in slugs using ¹³C labelling under light and dark conditions. In addition, they demonstrate that eggs produced by sea slugs under actinic light lay statistically significantly more eggs than those kept in non-actinic light. Taken together, these three lines of analysis strongly support the hypothesis that kleptoplasty provides nutrition for sea slugs, and improves their overall evolutionary fitness.

The author's major conclusions are:

1. Sequestered chloroplasts in *E. timida* actively photosynthesize fixing inorganic carbon to organic molecules which are then translocated to non-photosynthetic tissues in the animal involved with reproduction
2. Radiolabelled carbon was shown to be incorporated in a range of fatty acids, including those associated with reproduction, in animals exposed to light only. This experiment also provided an opportunity to trace the biochemistry of fatty acid synthesis in the animal.
3. Egg fecundity was statistically higher in slugs reared under actinic conditions vs non-actinic.

Comments: This is a very well written manuscript. The experiments are well designed, well executed, and conclusively support the hypothesis that *E. timida* rely on photosynthesis from

their stolen algal chloroplasts. While there is significant data in the literature supporting this position, recent papers have cast doubt on the role of photosynthesis. In my opinion, this study is of particular importance because it is the first to prove the link between photosynthesis and actual reproductive fitness in the animals. I recommend this manuscript for publication and only have a few minor comments.

Minor comments:

Line 121: You state that the ^{13}C level “generally leveled out”. Could you clarify what this means? As in the rate of incorporation leveled out, or the total amount leveled out indicating no new incorporation of ^{13}C ?

Figures: In general, I think more detailed descriptions of the figures would be helpful.

For example:

Fig 1: Those less adept at microscopy or slug anatomy may not be able to determine where the ^{13}C and ^{15}N are localized. It would be helpful to clarify in the figure text what is going on in the pictures.

Fig 2: The figures have the albumen glands labelled, but why not labels for the digestive tubule?

Fig 3: What is the tissue on the left hand side of the 3H chase micrograph? Are those follicles? Again, why not label more regions for clarity?

Referee: 2

Comments to the Author(s)

Several papers have demonstrated that kleptoplasts in the sea slug provide photosynthate and support growth, but recent study has denied it. Therefore, the role of kleptoplasts in the sea slug is still controversial. This study demonstrated that photosynthate from kleptoplasts are translocated to kleptoplasts-free reproductive organs and it might enhance the reproductive fitness. Authors showed the translocation of photosynthate by tracking short-term light-dependent incorporation of inorganic carbon (^{13}C) and nitrogen (^{15}N) into animal tissues using compound specific isotope analysis (CSIA) of fatty acid methyl esters (FAME) and high-resolution secondary ion mass spectrometry (NanoSIMS). This reviewer feels the experimental design is very reasonable and results are clear for this portion of the paper. However, authors also showed the physiological significance of the photosynthate from kleptoplasts by testing the effect of light on fecundity of sea slug. This reviewer feels the conclusion that the photosynthate from kleptoplasts is important for the reproductive fitness of the sea slug is not supported by the current data set. This reviewer feels the data reported in this paper is important to understanding the role of the kleptoplasts in sea slugs, however, in order to recommend publication either additional data needs to be added or the discussion/conclusion needs to be modified.

Major comment

In order to conclude that photosynthesis enhances the reproductive fitness of the sea slug, photosynthesis needed to be directly and specifically suppressed using an inhibitor such as DCMU, rather than reduced light intensity. The authors cannot exclude that reduced light intensity may directly reduce the fecundity of the sea slugs. Author need to add new data to support their conclusion. Alternatively, authors need to change this conclusion to discussion. Furthermore, throughout the text (and the title), it should be changed to say reduced light intensity and not inhibited/suppressed photosynthesis.

Minor comment

L62-65, It might be important to say that there is no horizontal gene transfer from macroalgae to sea slug (Maeda et al. 2021 eLife).

Author's Response to Decision Letter for (RSPB-2021-1331.R0)

See Appendix A.

RSPB-2021-1779.R0

Review form: Reviewer 1

Recommendation

Accept as is

Scientific importance: Is the manuscript an original and important contribution to its field?

Excellent

General interest: Is the paper of sufficient general interest?

Excellent

Quality of the paper: Is the overall quality of the paper suitable?

Excellent

Is the length of the paper justified?

Yes

Should the paper be seen by a specialist statistical reviewer?

No

Do you have any concerns about statistical analyses in this paper? If so, please specify them explicitly in your report.

No

It is a condition of publication that authors make their supporting data, code and materials available - either as supplementary material or hosted in an external repository. Please rate, if applicable, the supporting data on the following criteria.

Is it accessible?

Yes

Is it clear?

Yes

Is it adequate?

Yes

Do you have any ethical concerns with this paper?

No

Comments to the Author

My original review had only minor concerns which were adequately addressed in this revision. The majority of changes appear to be in response to the other reviewer. I feel that the author's adequately addressed these concerns as well.

Review form: Reviewer 2

Recommendation

Accept with minor revision (please list in comments)

Scientific importance: Is the manuscript an original and important contribution to its field?
Good

General interest: Is the paper of sufficient general interest?
Acceptable

Quality of the paper: Is the overall quality of the paper suitable?
Acceptable

Is the length of the paper justified?
Yes

Should the paper be seen by a specialist statistical reviewer?
No

Do you have any concerns about statistical analyses in this paper? If so, please specify them explicitly in your report.
No

It is a condition of publication that authors make their supporting data, code and materials available - either as supplementary material or hosted in an external repository. Please rate, if applicable, the supporting data on the following criteria.

Is it accessible?
N/A

Is it clear?
N/A

Is it adequate?
N/A

Do you have any ethical concerns with this paper?
No

Comments to the Author

The authors conclude that photosynthesis of kleptoplasts enhances the reproductive fitness of kleptoplasts-bearing sea slugs. However, this conclusion is not supported in current data set provided, as this reviewer previously commented. The authors chose to revise the manuscript according to this reviewer's comment. However, the revised manuscript still states that photosynthesis enhances the reproductive fitness of kleptoplasts-bearing sea slugs. Therefore, this reviewer feels the revisions are insufficient. Authors need to understand that reporting this kind of unsupported conclusion discourages further research.

Specific comments:

Title should be changed to something like "Photosynthesis of stolen chloroplast might support sea slug reproductive fitness".

In the abstract, "Finally, we report reduced fecundity of *E. timida* by limiting kleptoplasts photosynthesis" should be "Finally, we report reduced fecundity of *E. timida* in conditions where the kleptoplasts photosynthesis is limited".

Also in the abstract, "The present study provides the first thorough experimental evidence that photosynthesis enhances the reproductive fitness of kleptoplasts-bearing sea slugs, ..." is not appropriate. It should be "The present study suggests that photosynthesis enhances the reproductive fitness of kleptoplasts-bearing sea slugs, ...".

Lines 90-91, sentence should be "We report strong experimental data supporting a role of photosynthesis in the reproductive investment and fitness of a kleptoplast-bearing sea slug."

Lines 265-273, authors can suggest, but not conclude, that photosynthesis of kleptoplasts enhances the reproductive fitness of kleptoplasts-bearing sea slugs.

Decision letter (RSPB-2021-1779.R0)

03-Sep-2021

Dear Dr Cruz

I am pleased to inform you that your manuscript RSPB-2021-1779 entitled "Photosynthesis from stolen chloroplasts increases sea slug reproductive fitness" has been accepted for publication in Proceedings B.

The referee(s) have recommended publication, but also suggest some minor revisions to your manuscript. Therefore, I invite you to respond to the referee(s)' comments and revise your manuscript. Because the schedule for publication is very tight, it is a condition of publication that you submit the revised version of your manuscript within 7 days. If you do not think you will be able to meet this date please let us know.

Sincerely,

Dr Daniel Costa

Associate Editor

Comments to Author:

Thank you for revising and resubmitting your manuscript. Reviewer 1 is fully satisfied with your revisions, while Reviewer 2 suggests minor revisions. Reviewer 2's comments still revolve around the first comment made regarding strong statements on the direct link between the kleptoplast photosynthesis and reproductive fitness. In your revision, you toned down some of these statements as a response to Reviewer 2, particularly in your conclusion section. You could possibly similarly fine tune at least the title and the last sentence of the abstract for consistency. The other sentences pointed out though seem to be already fine as is.

Reviewer(s)' Comments to Author:

Referee: 1

Comments to the Author(s).

My original review had only minor concerns which were adequately addressed in this revision.

The majority of changes appear to be in response to the other reviewer. I feel that the author's adequately addressed these concerns as well.

Referee: 2

Comments to the Author(s).

The authors conclude that photosynthesis of kleptoplasts enhances the reproductive fitness of kleptoplasts-bearing sea slugs. However, this conclusion is not supported in current data set provided, as this reviewer previously commented. The authors chose to revise the manuscript according to this reviewer's comment. However, the revised manuscript still states that photosynthesis enhances the reproductive fitness of kleptoplasts-bearing sea slugs. Therefore, this reviewer feels the revisions are insufficient. Authors need to understand that reporting this kind of unsupported conclusion discourages further research.

Specific comments:

Title should be changed to something like "Photosynthesis of stolen chloroplast might support sea slug reproductive fitness".

In the abstract, "Finally, we report reduced fecundity of *E. timida* by limiting kleptoplasts photosynthesis" should be "Finally, we report reduced fecundity of *E. timida* in conditions where the kleptoplasts photosynthesis is limited".

Also in the abstract, "The present study provides the first thorough experimental evidence that photosynthesis enhances the reproductive fitness of kleptoplasts-bearing sea slugs, ..." is not appropriate. It should be "The present study suggests that photosynthesis enhances the reproductive fitness of kleptoplasts-bearing sea slugs, ...".

Lines 90-91, sentence should be "We report strong experimental data supporting a role of photosynthesis in the reproductive investment and fitness of a kleptoplast-bearing sea slug".

Lines 265-273, authors can suggest, but not conclude, that photosynthesis of kleptoplasts enhances the reproductive fitness of kleptoplasts-bearing sea slugs.

Author's Response to Decision Letter for (RSPB-2021-1779.R0)

See Appendix B.

Decision letter (RSPB-2021-1779.R1)

07-Sep-2021

Dear Dr Cruz

I am pleased to inform you that your manuscript entitled "Photosynthesis from stolen chloroplasts can support sea slug reproductive fitness" has been accepted for publication in Proceedings B.

Data Accessibility section

Open Access

Paper charges

Sincerely,

Appendix A

Comments to Reviewers

Reviewer 1:

Review of Manuscript ID: RSPB-2021-1331 Photosynthesis from stolen chloroplasts increases sea slug reproductive fitness

Summary: This is a strong paper which details careful experiments to demonstrate that the sea slug *Elysia timida* derives significant nutritional benefit from its kleptoplastic association with chloroplasts stolen from *Acetabularia acetabulum*. The authors use ^{13}C and ^{15}N labelling to show that organic molecules produced by photosynthesis are incorporated into the sea slug tissues, and translocated to the albumen gland and gonadal follicles (which are NOT located near chloroplasts). The use of NanoSIMS isotopic imaging was particularly useful in demonstrating this transfer. They also measured the amount and types of fatty acids, included those reported to be used in reproduction, in slugs using ^{13}C labelling under light and dark conditions. In addition, they demonstrate that eggs produced by sea slugs under actinic light lay statistically significantly more eggs than those kept in non-actinic light. Taken together, these three lines of analysis strongly support the hypothesis that kleptoplasty provides nutrition for sea slugs, and improves their overall evolutionary fitness.

The author's major conclusions are:

1. Sequestered chloroplasts in *E. timida* actively photosynthesize fixing inorganic carbon to organic molecules which are then translocated to non-photosynthetic tissues in the animal involved with reproduction
2. Radiolabelled carbon was shown to be incorporated in a range of fatty acids, including those associated with reproduction, in animals exposed to light only. This experiment also provided an opportunity to trace the biochemistry of fatty acid synthesis in the animal.
3. Egg fecundity was statistically higher in slugs reared under actinic conditions vs non-actinic.

Comments: This is a very well written manuscript. The experiments are well designed, well executed, and conclusively support the hypothesis that *E. timida* rely on photosynthesis from their stolen algal chloroplasts. While there is significant data in the literature supporting this position, recent papers have cast doubt on the role of photosynthesis. In my opinion, this study is of particular importance because it is the first to prove the link between photosynthesis and actual reproductive fitness in the animals. I recommend this manuscript for publication and only have a few minor comments.

Reply: The authors sincerely thank the positive feedback on our work by Reviewer 1.

Minor comments:

Line 121: You state that the ^{13}C level "generally leveled out". Could you clarify what this means? As in the rate of incorporation leveled out, or the total amount leveled out indicating no new incorporation of ^{13}C ?

Reply: We agree that the sentence is not clear. What we mean is that levels of ^{13}C stopped increasing, indicating no new incorporation. The sentence was rephrased for clarity. It now reads as follows: "In the chasing phase, when individuals were transferred to fresh non-labelled ASW, levels of ^{13}C in FA stopped increasing (Fig. 4)."

Figures: In general, I think more detailed descriptions of the figures would be helpful.

For example:

Fig 1: Those less adept at microscopy or slug anatomy may not be able to determine where the ^{13}C and ^{15}N are localized. It would be helpful to clarify in the figure text what is going on in the pictures.

Reply: A short explanation was added to the figure legend as suggested by the reviewer: “Digestive tubules (dt) appear green on the light microscopy micrographs due to the presence of numerous chloroplasts. After 6 hours of incubation strong ^{13}C and ^{15}N enrichment is observed in these structures”.

Fig 2: The figures have the albumen glands labelled, but why not labels for the digestive tubule?

Reply: Labels for the digestive tubules were added to Figure 2.

Fig 3: What is the tissue on the left hand side of the 3H chase micrograph? Are those follicles? Again, why not label more regions for clarity?

Reply: The structure in Fig. 3H is a gonadal follicle, as labelled. There is also a bit of digestive tubule at the lower left side. Other labels for the identified structures were added to Figure 3 for clarity.

Reviewer 2:

Several papers have demonstrated that kleptoplasts in the sea slug provide photosynthate and support growth, but recent study has denied it. Therefore, the role of kleptoplasts in the sea slug is still controversial. This study demonstrated that photosynthate from kleptoplasts are translocated to kleptoplast-free reproductive organs and it might enhance the reproductive fitness. Authors showed the translocation of photosynthate by tracking short-term light-dependent incorporation of inorganic carbon (^{13}C) and nitrogen (^{15}N) into animal tissues using compound specific isotope analysis (CSIA) of fatty acid methyl esters (FAME) and high-resolution secondary ion mass spectrometry (NanoSIMS). This reviewer feels the experimental design is very reasonable and results are clear for this portion of the paper. However, authors also showed the physiological significance of the photosynthate from kleptoplasts by testing the effect of light on fecundity of sea slug. This reviewer feels the conclusion that the photosynthate from kleptoplasts is important for the reproductive fitness of the sea slug is not supported by the current data set. This reviewer feels the data reported in this paper is important to understanding the role of the kleptoplasts in sea slugs, however, in order to recommend publication either additional data needs to be added or the discussion/conclusion needs to be modified.

Reply: We would like to clarify that our conclusion that kleptoplast photosynthesis increases sea slug reproductive fitness is based on three lines of evidence: i) light-dependent incorporation of ^{13}C and ^{15}N in the albumen gland and gonadal follicles, representing translocation of photosynthates to kleptoplast-free reproductive organs; ii) production of long-chain polyunsaturated fatty acids with reported roles in reproduction in the sea slug cells using

labelled precursors translocated from the kleptoplasts; and iii) higher fecundity in sea slugs reared under regular light than in quasi-dark conditions. The reviewer does not question points i and ii, but has some concerns regarding higher fecundity described in point iii being directly related to kleptoplast photosynthesis. We address these concerns below.

Major comment

In order to conclude that photosynthesis enhances the reproductive fitness of the sea slug, photosynthesis needed to be directly and specifically suppressed using an inhibitor such as DCMU, rather than reduced light intensity. The authors cannot exclude that reduced light intensity may directly reduce the fecundity of the sea slugs. Author need to add new data to support their conclusion. Alternatively, authors need to change this conclusion to discussion. Furthermore, throughout the text (and the title), it should be changed to say reduced light intensity and not inhibited/suppressed photosynthesis.

Reply: If we use of a chemical blocker of photosynthesis as suggested by the reviewer one could argue that the blocker could directly reduce the fecundity of the sea slugs, as the reviewer argues for reduced light. It is unrealistic to think that an herbicide like DCMU (or monolinuron as used by Christa et al. 2014, Proc R Soc B 281, 20132493) would have no effect on animal metabolism when used for so long on the sea slugs (in our case 28 days). In fact, we have strong experimental evidence that monolinuron strongly affects neurotransmission and oxidative stress response in sea slugs in as short as 4 days of exposure to concentrations of 2 mg/L (as the one used by Christa et al. 2014 and in other works of the same research group to block photosynthesis of sea slugs). Furthermore, it would require the slugs to be reared in much smaller volumes of water than in the 150 L life support systems used in our study. For example, Rauch et al. 2018 (Mar Biol 165:82) used petri dishes to maintain sea slugs for 21 days exposed to a photosynthesis blocker, which is completely inadequate with severe impact on animals well-being and promoting unreliable results. Finally, as recognized by Christa et al. 2014 (J Molluscan Stud 80: 499–507), these blockers show only partial inhibition of kleptoplast photosynthesis in sea slugs. By the reasons presented above, we chose the reviewers' alternative to change the text to accommodate his concerns. We now only refer that sea slug fecundity was statistically higher in slugs reared under regular light conditions and that we hypothesize that this is due to kleptoplast photosynthesis. It now reads as follows: "Although we cannot rule out a direct effect of experimental light conditions in sea slug spawning, we hypothesize that observed differences in fecundity were due to resources provided by kleptoplast photosynthesis." The use of very low light instead of darkness not to affect animal behavior is discussed in the text and used before to limit photosynthesis in sea slugs (e.g. Shiroyama et al. 2020 Oecologia 194, 455-463; Baumgartner et al. 2015 PLoS ONE 10, e0120874). As suggested by the reviewer, we also changed the term "inhibited/suppressed photosynthesis" to "limited photosynthesis" and the terms "actinic" and "non-actinic" to "regular light" and "reduced light" conditions. The conclusion paragraph was changed accordingly. We hope that the reviewer considers that on its revised form our work is now suitable for publication in Proc R Soc B.

Minor comment

L62-65, It might be important to say that there is no horizontal gene transfer from macroalgae to sea slug (Maeda et al. 2021 eLife).

Reply: A sentence was added to the text on the absence of horizontal gene transfer in photosynthetic sea slugs with reference to Maeda et al. 2021, as suggested by the reviewer: “Although horizontal gene transfer was suggested as the primary reason underlying long-term maintenance of photosynthesis in sacoglossan sea slugs, more recent studies found no evidence of genes supporting photosynthesis in the animal nuclear DNA (Maeda et al. 2021)”.

Appendix B

Comments to Reviewers

Reviewer 1:

Comments to the Author(s).

My original review had only minor concerns which were adequately addressed in this revision. The majority of changes appear to be in response to the other reviewer. I feel that the author's adequately addressed these concerns as well.

Reply: The authors sincerely thank Reviewer 1 for previous suggestions to improve our manuscript.

Reviewer 2:

Comments to the Author(s).

The authors conclude that photosynthesis of kleptoplasts enhances the reproductive fitness of kleptoplasts-bearing sea slugs. However, this conclusion is not supported in current data set provided, as this reviewer previously commented. The authors chose to revise the manuscript according to this reviewer's comment. However, the revised manuscript still states that photosynthesis enhances the reproductive fitness of kleptoplasts-bearing sea slugs. Therefore, this reviewer feels the revisions are insufficient. Authors need to understand that reporting this kind of unsupported conclusion discourages further research.

Reply: Our conclusion that kleptoplast photosynthesis can support sea slug reproductive fitness is based on three lines of evidence presented: i) light-dependent incorporation of ¹³C and ¹⁵N in the albumen gland and gonadal follicles, representing translocation of photosynthates to kleptoplast-free reproductive organs; ii) production of long-chain polyunsaturated fatty acids with reported roles in reproduction in the sea slug cells using labelled precursors translocated from the kleptoplasts; and iii) higher fecundity in sea slugs reared under regular light than in quasi-dark conditions. We believe the conclusion is not unsupported and that it does not discourage further research.

Specific comments:

Title should be changed to something like "Photosynthesis of stolen chloroplast might support sea slug reproductive fitness".

Reply: The title was changed accordingly to "Photosynthesis of stolen chloroplast can support sea slug reproductive fitness".

In the abstract, "Finally, we report reduced fecundity of *E. timida* by limiting kleptoplasts photosynthesis" should be "Finally, we report reduced fecundity of *E. timida* in conditions where the kleptoplasts photosynthesis is limited".

Reply: The sentence suggested has the same meaning.

Also in the abstract, “The present study provides the first thorough experimental evidence that photosynthesis enhances the reproductive fitness of kleptoplasts-bearing sea slugs, ...” is not appropriate. It should be “The present study suggests that photosynthesis enhances the reproductive fitness of kleptoplasts-bearing sea slugs, ...”.

Reply: The sentence was changed accordingly to: “The present study indicates that photosynthesis enhances the reproductive fitness of kleptoplast-bearing sea slugs...”

Lines 90-91, sentence should be “We report strong experimental data supporting a role of photosynthesis in the reproductive investment and fitness of a kleptoplast-bearing sea slug.”.

Reply: The sentence suggested has the same meaning.

Lines 265-273, authors can suggest, but not conclude, that photosynthesis of kleptoplasts enhances the reproductive fitness of kleptoplasts-bearing sea slugs.

Reply: This paragraph was already revised in the previous version, changing the words “demonstrate” to “indicate” and “show” to “hypothesize”.